# Clinicopathological Features of Small Pancreatic Neuroendocrine Neoplasms 10 mm or Smaller

**DOI:** 10.3390/diagnostics15192423

**Published:** 2025-09-23

**Authors:** Tetsuro Hirano, Masahiro Serikawa, Yasutaka Ishii, Shinya Nakamura, Juri Ikemoto, Masaru Furukawa, Yumiko Yamashita, Noriaki Iijima, Yasuhiro Okuda, Risa Nomura, Koji Arihiro, Kenichiro Uemura, Shinya Takahashi, Hideki Ohdan, Shiro Oka

**Affiliations:** 1Department of Gastroenterology, Graduate School of Biomedical and Health Sciences, Hiroshima University, Hiroshima 734-8551, Japan; techandarky44@gmail.com (T.H.);; 2Department of Gastroenterology, Onomichi General Hospital, Hiroshima 722-8508, Japan; 3Department of Gastroenterology, Hiroshima Prefectural Hospital, Hiroshima 734-8530, Japan; 4Department of Anatomical Pathology, Hiroshima University Hospital, Hiroshima 734-8551, Japan; 5Department of Surgery, Graduate School of Biomedical & Health Sciences, Hiroshima University, Hiroshima 734-8551, Japan; 6Department of Gastroenterological and Transplant Surgery, Graduate School of Biomedical & Health Sciences, Hiroshima University, Hiroshima 734-8551, Japan

**Keywords:** pancreatic neuroendocrine neoplasm, imaging, endoscopic ultrasound-guided fine needle aspiration, pathological diagnosis, malignancy

## Abstract

**Background/Objectives:** There is limited evidence on imaging characteristics and pathological grading of small pancreatic neuroendocrine tumors (PNENs). This study aimed to compare imaging features and histopathological diagnoses of PNENs based on tumor size. **Methods:** Patients with PNEN who underwent pathological diagnosis at Hiroshima University Hospital were retrospectively reviewed. Detection rates, imaging findings, and diagnostic accuracy of endoscopic ultrasound-guided fine needle aspiration (EUS-FNA) were analyzed according to tumor size. **Results:** Among 107 patients with PNEN, 42 had tumors ≤10 mm, and 65 had tumors >10 mm. The detection rate for lesions ≤10 mm was 92.3% according to EUS and 83.3% according to contrast-enhanced CT, showing the superior sensitivity of EUS. Typical imaging features—well-defined margins, homogeneous internal structure, and early enhancement—were significantly more frequent in tumors ≤10 mm (*p* < 0.001). There were no significant differences in imaging findings between G1 and G2 tumors ≤10 mm. The diagnostic sensitivity of EUS-FNA was 91.2% for tumors ≤10 mm and 86.3% for tumors >10 mm, with no significant difference. However, the concordance rate between EUS-FNA and surgical histology for tumor grading was significantly higher in the ≤10 mm group (87.5% vs. 56.3%, *p* = 0.012). **Discussion:** In small PNENs (≤10 mm), imaging features are often typical but do not reliably determine tumor grade. In our cohort, EUS-FNA showed high diagnostic accuracy and provided essential pathological information to guide management, including the choice between surveillance and surgery.

## 1. Introduction

In recent years, advances in abdominal ultrasound, computed tomography (CT), magnetic resonance imaging (MRI), and endoscopic ultrasound (EUS) have increased the incidental detection of small pancreatic neuroendocrine neoplasms (PNENs) [1,2]. Current clinical guidelines recommend treatment strategies for non-functional PNENs (NF-PNENs) mainly according to tumor size, with >20 mm generally used as the threshold for surgical resection because larger tumors are associated with higher malignant potential [3,4]. However, the optimal management of small NF-PNENs remains controversial [5,6,7,8].

For incidentally discovered NF-PNENs ≤10 mm, active surveillance is often considered preferable to surgery, given the risks of perioperative complications and long-term sequelae [9,10,11]. Nonetheless, several studies have shown that even tumors ≤10 mm may harbor malignant potential, including lymph node metastasis, particularly when atypical imaging features are present or when the tumor is classified as G2 [12]. These findings suggest that tumor size alone is not sufficient for reliable prognostic assessment. Instead, accurate pathological grading, especially based on the Ki-67 index, is essential to guiding treatment decisions.

Endoscopic ultrasound-guided fine needle aspiration (EUS-FNA) is the most widely used method for preoperative pathological diagnosis of PNENs, but its diagnostic accuracy in very small tumors (≤10 mm) remains unclear. Concerns have been raised regarding technical feasibility, sample adequacy, and the reliability of grading in such small lesions [13,14,15]. Moreover, few studies have directly compared imaging findings with histopathological grade in this subgroup.

We hypothesized that EUS-FNA provides more accurate pathological grading for small PNENs (≤10 mm) than for larger tumors, and that imaging findings alone cannot reliably distinguish between G1 and G2 tumors of this size. The primary aim of this study was to evaluate the diagnostic accuracy of EUS-FNA and to compare imaging and pathological features according to tumor size. By clarifying these aspects, our study contributes to optimizing clinical decision making regarding surveillance versus surgical resection for small PNENs.

## 2. Materials and Methods

### 2.1. Participants

From April 2011 to March 2024, a total of 109 patients were diagnosed with PNENs at Hiroshima University Hospital. Among them, 107 patients who underwent histopathological confirmation of tumor malignancy through either EUS-FNA or surgical specimens were included in this study. Two patients who underwent imaging studies but did not undergo pathological examination were excluded (Figure 1).

At our institution, even small PNENs (≤10 mm) were surgically resected in selected cases. Resection was considered when imaging findings were atypical, when patients preferred definitive diagnosis or treatment, or when histological confirmation was required. During the earlier years of the study period (2011–2024), clinical guidance on the management of small PNENs was evolving, and a standardized watch-and-wait approach had not yet been widely implemented in routine practice.

This study was approved by the Ethics Committee of the Graduate School of Biomedical and Health Sciences, Hiroshima University (Hiroshima, Japan) (Approval No.: E-519, 16 June 2017). Owing to the retrospective design of the study, the requirement for informed consent was waived.

### 2.2. Endpoints

The primary endpoints of this study were to assess the adequacy of tissue sampling and the diagnostic accuracy of EUS-FNA for small PNENs by comparing malignancy classifications obtained from EUS-FNA with final pathological evaluations from surgical specimens. The secondary endpoints were to analyze detailed imaging features of small PNENs and evaluate their correlation with malignancy, also using final surgical pathology as the reference standard.

### 2.3. Image Acquisition

All participants underwent multi-detector row computed tomography (MDCT). For contrast-enhanced CT, dynamic studies were conducted using either a four-phase protocol (non-contrast, late arterial, portal venous, and equilibrium phases) or a five-phase protocol (non-contrast, early arterial, pancreatic parenchymal, portal venous, and equilibrium phases). Imaging timings after intravenous administration of 100 mL of non-ionic contrast agent were as follows: early arterial phase at 25 s, late arterial phase at 40 s, pancreatic parenchymal phase at 50 s, portal venous phase at 70 or 75 s, and equilibrium phase at 100 or 120 s.

EUS was performed using either a radial scanning endoscope (GF-UE260-AL5, GF-UE290; Olympus Medical Systems, Tokyo, Japan) or a convex scanning endoscope (GF-UCT260; Olympus Medical Systems) connected to an ultrasound platform (Prosound SSDα-10; Aloka Co., Ltd., Tokyo, Japan, or EU-ME1/EU-ME2; Olympus Medical Systems). All EUS examinations were conducted at 7.5 MHz by experienced endoscopists with a minimum of 5 years of experience.

### 2.4. Image Analysis

The following imaging features were evaluated (Figure 2):(a)tumor margin contour on EUS (regular or irregular),(b)internal echogenicity on EUS (homogeneous or heterogeneous),(c)arterial phase enhancement pattern on contrast-enhanced CT (presence or absence of early enhancement),(d)presence of calcifications on CT or EUS (identified as hyperechoic structures or high-attenuation areas),(e)presence of cystic degeneration on CT or EUS (non-enhancing anechoic areas),(f)main pancreatic duct (MPD) dilation (MPD diameter ≥3 mm on CT or EUS).

Definitions of imaging characteristics:Irregular tumor margins: tumors with spiculated or infiltrative borders in ≥20% of the lesion circumference.Heterogeneous echogenicity: tumors with internal echo variability not attributable to cystic changes or calcifications.Early enhancement: tumors show greater contrast enhancement than the normal pancreatic parenchyma during the arterial phase.Calcifications: hyperechoic areas on EUS correspond to high attenuation on non-contrast CT.Cystic degeneration: anechoic regions on EUS without enhancement on any contrast-enhanced CT phase.MPD dilation: maximal MPD diameter ≥3 mm proximal to the index lesion on CT or EUS [16,17].

A “typical imaging case” was defined as one meeting all six of the following criteria to ensure high specificity and minimize misclassification across heterogeneous imaging protocols:

(1) regular margins, (2) homogeneous structure, (3) early enhancement present, (4) no calcifications, (5) no cystic degeneration, and (6) no MPD dilation.

An “atypical imaging case” was defined as one meeting at least one of the following criteria:

(1) irregular margins, (2) heterogeneous structure, (3) absence of early enhancement, (4) presence of calcifications, (5) presence of cystic degeneration, or (6) presence of MPD dilation.

### 2.5. EUS-FNA Procedures

EUS-FNA was performed using a convex scanning echoendoscope (GF-UCT260; Olympus Medical Systems). The needles used included 19G, 22G, and 25G types from various manufacturers: Acquire and Expect (Boston Scientific Corp., Marlborough, MA, USA), EZ Shot 3 Plus (Olympus Medical Systems), SonoTip Pro Control (Medico’s Hirata Inc., Osaka, Japan), and EchoTip ProCore (Cook Medical Japan G.K., Tokyo, Japan). The choice between FNA and FNB needles, as well as the needle gauge (19G or 22G or 25G), was at the discretion of the endoscopist based on lesion characteristics and procedural stability. FNA was more commonly used in the earlier years of the study period, and the use of FNB increased in the later years with the broader availability of core needles and growing evidence, particularly when architectural preservation was required or when a PNEN was suspected.

Tissue samples were obtained either using aspiration with a 20 mL syringe or via the slow-pull technique without negative pressure, also selected at the endoscopist’s discretion. Rapid On-Site Evaluation (ROSE) was available throughout the study period and was performed when a cytopathologist or trained cytotechnologist was present during the procedure. Adverse events related to EUS-FNA were monitored within 7 days using medical records, laboratory results, and imaging, and severity was to be classified according to the American Society for Gastrointestinal Endoscopy lexicon for endoscopic adverse events [18].

All procedures were performed by experienced endoscopists with at least 5 years of expertise in EUS-FNA.

### 2.6. Histopathological Evaluation

Histopathological diagnoses were made using tissue samples obtained through EUS-FNA. In patients who underwent surgical resection, the final diagnosis of tumor malignancy was based on the examination of surgical specimens. Tumors were graded according to the 2022 World Health Organization classification [19], which categorizes neuroendocrine neoplasms into grade 1 (NET G1), grade 2 (NET G2), grade 3 (NET G3), and neuroendocrine carcinoma (NEC). For patients who had been diagnosed prior to the publication of the 2022 classification, all specimens were re-evaluated and reclassified by experienced pathologists according to the updated criteria. All histopathological assessments were performed by at least two experienced pathologists. Although information on the suspected diagnosis of pancreatic neuroendocrine neoplasm was available, the evaluations were conducted independently of detailed clinical and imaging findings. Representative histopathological images are presented in Appendix A.

### 2.7. Use of AI-Assisted Technologies

During the preparation of this work, the authors used ChatGPT (GPT-5 Thinking, OpenAI, San Francisco, CA, USA) to assist with translation and improve the grammar, clarity, and accuracy of the English text. After using this tool/service, the authors reviewed and edited the content as needed and take full responsibility for the content of the publication.

### 2.8. Statistical Analysis

All statistical analyses were performed using JMP Pro 18 (SAS Institute Inc., Cary, NC, USA). Continuous variables were compared using the Wilcoxon rank-sum test, while categorical variables were analyzed using the chi-square test or Fisher’s exact test, as appropriate. An atypicality score from 0 to 6 (higher values indicate more atypical appearance) was constructed by summing six predefined imaging features. The score was modeled as an ordinal outcome using proportional odds logistic regression. Lesion size was rescaled in 10 mm units, and odds ratios per 10 mm increase with 95 percent confidence intervals (CIs) were reported. For the primary comparison between ≤10 mm (*n* = 42) and >10 mm (*n* = 65), a post hoc minimum detectable absolute difference at alpha = 0.05 and power = 0.80 was computed, yielding approximately 24–27 percentage points across baseline proportions of 20 to 40 percent in the reference group. A two-tailed *p*-value < 0.05 was considered statistically significant.

## 3. Results

### 3.1. Patient Characteristics

Patient characteristics and clinical comparisons based on tumor diameter are summarized in Table 1. Among the 107 patients, 42 had tumors measuring ≤10 mm, and 65 had tumors >10 mm in diameter. Tumor location in the ≤10 mm group was distributed as follows: head (*n* = 10), body (*n* = 17), and tail (*n* = 15). In the >10 mm group, the distribution was head (*n* = 26), body (*n* = 14), and tail (*n* = 25). Tumors ≤10 mm were significantly more likely to be located in the body of the pancreas (*p* = 0.035). Histopathological grades of malignancy in the ≤10 mm group included 29 patients of G1 and 13 patients of G2, with no patients of either G3 or NEC. In contrast, the >10 mm group comprised 29 patients of G1, 27 of G2, 4 of G3, and 5 of NEC. The proportion of G1 tumors was significantly higher in the ≤10 mm group (*p* = 0.013). EUS-FNA was performed in 85 patients (34 in the ≤10 mm group and 51 in the >10 mm group), while surgical resection was conducted in 92 patients (38 in the ≤10 mm group and 54 in the >10 mm group), with no significant difference between groups (*p* = 0.282).

### 3.2. Detection Rate of PNENs Using EUS and CT

Detection rates of PNENs by EUS and CT according to tumor size are shown in Table 2. EUS successfully detected all lesions >10 mm (100%, 62/62) and 92.3% (36/39) of lesions ≤10 mm (*p* = 0.027). The three lesions ≤10 mm that were not detected by EUS measured 1 mm, 8 mm, and 5 mm, respectively. CT detected 96.9% (62/64) of tumors >10 mm and 83.3% (35/42) of tumors ≤10 mm (*p* = 0.014). While CT showed good overall detection ability for larger tumors, EUS demonstrated superior detection performance for tumors ≤10 mm.

### 3.3. Comparison of Imaging Findings Between PNENs ≤10 mm and >10 mm

Table 3 presents the results of comparison of imaging findings that could be identified using both EUS and contrast-enhanced CT, divided into two groups based on tumor size: ≤10 mm and >10 mm (≤10 mm: *n* = 36, >10 mm: *n* = 61).

Among patients with tumors >10 mm, 26.2% (16/61) were classified as having typical imaging findings, while 73.8% (45/61) exhibited atypical features. In contrast, among those with tumors ≤10 mm, 72.2% (26/36) were typical cases, and 27.8% (10/36) were atypical. The proportion of typical imaging cases was significantly higher in the ≤10 mm group (*p* < 0.001).

When comparing individual imaging features, irregular tumor margins were observed in 8.3% of cases in the ≤10 mm group and 34.4% in the >10 mm group (*p* = 0.004). Heterogeneous internal structures were found in 2.8% and 44.3% of the ≤10 mm and >10 mm groups, respectively (*p* < 0.001). The absence of early enhancement was noted in 11.1% of tumors ≤10 mm and 42.6% of tumors >10 mm (*p* = 0.001). Calcifications within the tumor were present in 2.8% of tumors ≤10 mm and 19.7% of those >10 mm (*p* = 0.018). Cystic degeneration was detected in 5.6% of tumors ≤10 mm and 21.3% of tumors >10 mm (*p* = 0.038). Dilation of the MPD was found in 0% of tumors ≤10 mm and 23.0% of tumors >10 mm (*p* = 0.002). The frequency of atypical imaging features increased significantly with tumor size for all evaluated parameters. Size-stratified imaging features are visualized in Appendix A, complementing Table 3 and facilitating comparison between ≤10 mm and >10 mm. Lesion size was associated with higher atypicality categories. The odds ratio per 10 mm increase was 0.498 (95% CI 0.374–0.646; *p* < 0.001), indicating a shift toward more atypical features with increasing size. In sensitivity analyses excluding functioning tumors and/or MEN1 cases, the direction and magnitude of the associations were materially unchanged, with significant between-group differences persisting for most imaging features (Appendix A).

### 3.4. Comparison of Imaging Findings Between G1 and G2 PNENs ≤10 mm

Table 4 shows the comparison of imaging features between G1 and G2 tumors ≤10 mm in size, in cases where the lesions could be identified using both EUS and contrast-enhanced CT (G1: *n* = 25, G2: *n* = 11).

Atypical imaging features were observed in 28.0% (7/25) of G1 patients and 27.3% (3/11) of G2 patients, with no significant difference between the two groups (*p* = 0.964). Similarly, there were no significant differences in any specific imaging findings between G1 and G2 tumors in this size category.

Although these results indicate that imaging findings alone may not reliably distinguish between G1 and G2 tumors within the ≤10 mm group, we conducted an additional analysis that included all tumor sizes to examine imaging features across different pathological grades. As summarized in Table 5, the frequency of atypical imaging findings, including irregular margins, heterogeneous echogenicity, absence of early enhancement, and dilation of the main pancreatic duct, was higher in tumors with higher pathological grade. Typical imaging was observed in 51.9% of G1 tumors, 40.5% of G2 tumors, and none of the G3/NEC tumors (*p* = 0.020). Irregular margins and heterogeneous structures were found in 87.5% of G3/NEC tumors, compared to 13.5% and 15.4% in G1, and 27.0% and 35.1% in G2, respectively (*p* < 0.001 for both). Absence of early enhancement and MPD dilation were also more frequent in higher-grade tumors (*p* < 0.001 for both).

### 3.5. Diagnostic Accuracy of EUS-FNA According to Tumor Size

Table 6 compares the diagnostic accuracy of EUS-FNA based on tumor size. The technical success rate of EUS-FNA was 100% in both groups. No significant differences were observed between the ≤10 mm and >10 mm groups in terms of successful histopathological diagnosis of PNENs or determination of tumor grade. However, the concordance rate between malignancy grade determined by EUS-FNA and that determined from surgical specimens was significantly higher in the ≤10 mm group (87.5%, 21/24) than in the >10 mm group (56.3%, 18/32) (*p* = 0.012). No procedure-related complications occurred among the 85 patients who underwent EUS-FNA, yielding per-patient and per-procedure adverse event rates of 0%; no severity grading applied.

## 4. Discussion

PNENs were historically considered relatively rare tumors. However, with the advancement of imaging technology and growing clinical awareness, incidentally detected small non-functional PNENs are increasingly being identified. Consequently, appropriate strategies for early diagnosis and management have become important clinical issues. Accurate tumor detection through imaging and reliable pathological assessment of tumor grade are essential for selecting the most appropriate treatment approach.

CT and EUS are central imaging modalities used in the evaluation of PNENs. EUS, in particular, offers high spatial resolution and is less affected by gastrointestinal contents or subcutaneous fat, making it especially effective for detecting small pancreatic tumors. Previous reports have demonstrated that EUS has higher sensitivity than CT for detecting small PNENs. Manta et al. [20] found that CT failed to detect 68.4% of PNENs ≤10 mm and 15% of those ≤20 mm in diameter. Kurita et al. [21] compared detection rates of EUS and CT by tumor size, reporting EUS detection rates of 100% for tumors >20 mm, 95% for those 10–20 mm, 97.7% for 5–10 mm, and 58.4% for <5 mm; CT detection rates for the same size categories were 100%, 90%, 79.5%, and 16.7%, respectively. In line with these findings, the present study showed that while CT was effective for detecting tumors >10 mm, EUS demonstrated superior detection performance for lesions ≤10 mm. These results support the combined use of EUS and CT, especially for reliable detection of small PNENs.

The imaging features of PNENs are notably diverse, and atypical findings are frequently observed. Several recent studies have suggested that certain imaging characteristics may correlate with tumor grade, indicating the potential for noninvasive estimation of malignancy [12,22,23]. Typical features are generally defined as round shape, homogeneous internal structure, and early contrast enhancement, while atypical features include irregular margins, heterogeneous structure, lack of early enhancement, calcifications, cystic degeneration, and dilation of the MPD. In our analysis including tumors of all sizes, we observed a clear trend toward a higher frequency of atypical imaging features with higher tumor grade. Typical imaging was present in 51.9% of G1 tumors, 40.5% of G2 tumors, and none of the G3/NEC tumors, while irregular margins, heterogeneous echotexture, absence of early enhancement, and MPD dilation were significantly more frequent in G3/NEC tumors. These findings support an association between imaging appearance and pathological grade.

However, our size-stratified analysis revealed that such a correlation may not hold true for tumors ≤10 mm. As illustrated in Figure 3, both lesions were ≤10 mm and exhibited typical imaging features, yet differed in histopathological grade (G1 and G2), underscoring the difficulty of estimating tumor grade based on imaging alone in small PNENs. These observations suggest that morphological differences by tumor grade may not be sufficiently pronounced in small lesions to be reliably detected through imaging.

Another aim of this study was to assess whether the diagnostic accuracy of EUS-FNA for determining tumor grade varies by tumor size. EUS-FNA is widely regarded as the most reliable preoperative method for assessing tumor malignancy. Previous cohort studies have reported diagnostic accuracies ranging from 58% to 89% [13,14,15]. A prospective multicenter study by Kamata et al. using a 25-gauge core biopsy needle reported a 100% technical success rate, a 90.3% histological diagnosis rate, and an 82.6% concordance rate between EUS-FNA and surgical histopathology [24]. However, few studies have specifically addressed whether tumor size affects the diagnostic accuracy of EUS-FNA. Underestimation may occur when EUS-FNA classifies a tumor as G1, while surgical pathology reveals G2, potentially leading to inappropriate conservative management. Conversely, overestimation, in which a tumor is classified as G2 by EUS-FNA but is actually G1, may result in overtreatment. Tacelli et al. [25] reported that among tumors ≤20 mm, 22.7% were underestimated, and only 4.5% were overestimated in grade. Possible causes include contamination with proliferative non-tumor cells (e.g., crypt epithelium) or sampling of non-representative areas, particularly in small lesions. In our study, both the technical success rate and histological validity of EUS-FNA were high regardless of tumor size. However, the concordance rate of tumor grading between EUS-FNA and surgical specimens was significantly higher in tumors ≤10 mm (87.5%) than in those >10 mm (56.3%). This may be attributable to the relatively uniform cell distribution and preserved histoarchitecture in small tumors, which minimize sampling error. In contrast, larger PNENs are more likely to exhibit intratumoral necrosis, hemorrhage, fibrosis, or Ki-67 heterogeneity (so-called “hot spots”), which may compromise the reliability of small biopsy samples.

Sampling technique may influence diagnostic outcomes in small PNENs. Although our study did not stratify results by aspiration methods, prior evidence indicates that technique affects tissue adequacy and integrity. A recent network meta-analysis reported that the modified wet-suction technique yielded the highest sample adequacy and tissue integrity, though it was associated with higher rates of blood contamination [26]. A multicenter randomized crossover trial demonstrated that wet-suction achieved higher tissue core procurement and greater tissue integrity compared with the slow-pull technique, while diagnostic accuracy and tumor fraction were similar between the two methods [27]. These findings suggest that differences in sampling technique could partly explain variability in reported outcomes for small PNENs and should be considered in both clinical practice and future research.

Although current guidelines [3,4] recommend surveillance for non-functioning PNETs ≤10 mm, many of the cases included in our study were diagnosed and treated before these recommendations were widely established. During that time, surgical resection was often performed based on factors such as tumor growth, patient preference, the presence of atypical imaging features, or institutional protocols. This historical context should be taken into account when interpreting our findings.

This study has several important limitations. First, it was a retrospective study conducted at a single center. Second, the sample size, particularly for some subgroup analyses, is modest, which may limit external generalizability and statistical power to detect subtle differences or associations. Third, differences in needle selection or operator expertise during EUS-FNA may have affected the accuracy of histological grading. Fourth, although surgical specimens were used as the pathological gold standard for tumor grading, intratumoral heterogeneity in Ki-67 labeling may have led to sampling bias, even in resected tumors.

In conclusion, imaging alone is insufficient for determining tumor grade in small PNENs, and even small lesions may harbor malignant potential. EUS-FNA provides a reliable method for preoperative pathological diagnosis, and, in our study, it demonstrated high concordance with surgical specimen in tumors ≤10 mm. Our findings highlight the utility of EUS-FNA for assessing tumor grade, particularly in small lesions. However, clinical decisions regarding surgical resection should be made in conjunction with guideline recommendations, imaging features, patient factors, and institutional practices. In this context, our results underscore the importance of obtaining pathological confirmation even for small PNENs and reinforce the need for careful diagnostic and therapeutic decision-making in their management.

### Future Directions and Clinical Implications

The present findings indicate that imaging typicality alone is insufficient for reliable grading of very small PNENs and that pathological assessment obtained by EUS-FNA contributes meaningfully to size-stratified management, including the choice between surveillance and surgery. In clinical pathways, EUS-FNA may be prioritized for lesions 10 mm or smaller when atypical imaging features are present or when grade determination would alter treatment. Future work should validate the graded atypicality framework in multicenter cohorts using standardized sampling techniques and prespecified analytic plans, and evaluate prospective triggers for surveillance or intervention anchored to pathological grade. These steps could help translate the current evidence into practical algorithms while minimizing both overtreatment and undertreatment.

## Figures and Tables

**Figure 1 diagnostics-15-02423-f001:**
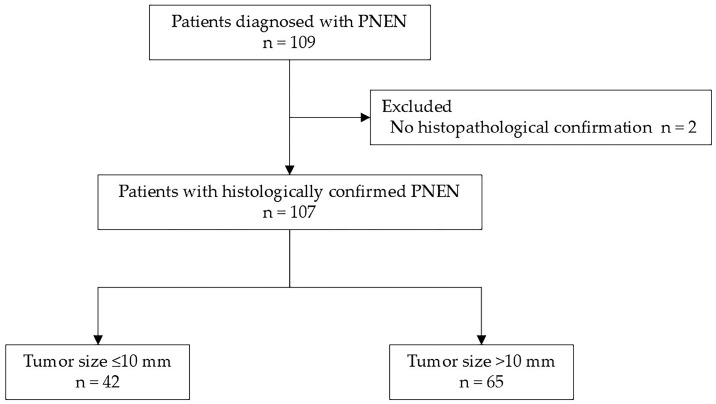
Flowchart of patient selection. Among 109 patients diagnosed with PNEN at our institution, 2 patients who underwent only imaging studies without histopathological confirmation were excluded. The remaining 107 patients were included in the analysis. Of these, 42 had tumors ≤10 mm in diameter, and 65 had tumors >10 mm. PNEN, pancreatic neuroendocrine neoplasm.

**Figure 2 diagnostics-15-02423-f002:**
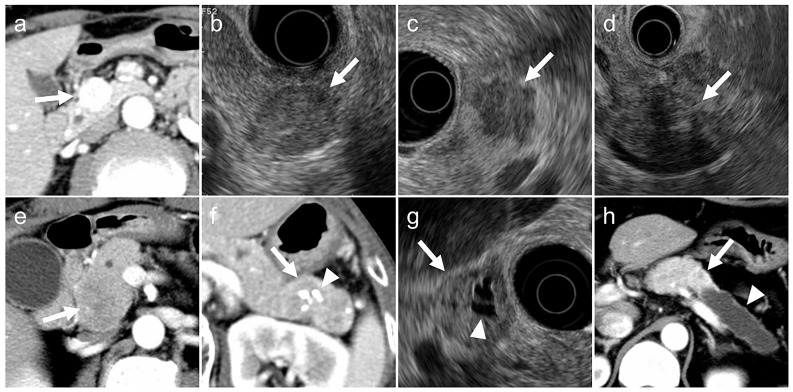
Typical (**a**,**b**) and atypical (**c**–**h**) imaging findings for pancreatic neuroendocrine neoplasms on contrast-enhanced computed tomography (CT) and endoscopic ultrasonography (EUS). (**a**). Arterial-phase CT showing a tumor with early enhancement (arrow). (**b**). EUS showing a tumor with smooth margins and homogeneous internal echotexture (arrow). (**c**). EUS showing a tumor with irregular margins (arrow). (**d**). EUS showing a tumor with internal heterogeneity (arrow). (**e**). Arterial-phase CT showing a tumor without early enhancement (arrow). (**f**). CT showing intratumoral calcifications (arrowhead) within the tumor (arrow). (**g**). EUS showing a tumor (arrow) with cystic degeneration (arrowhead). (**h**). CT showing a tumor (arrow) with main pancreatic duct dilatation (arrowhead).

**Figure 3 diagnostics-15-02423-f003:**
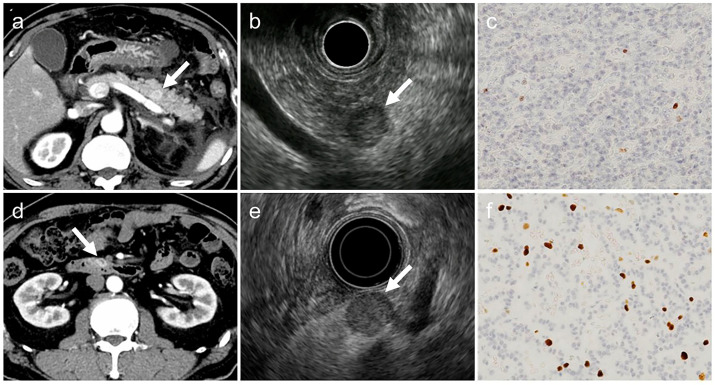
Two typical imaging cases in the group with tumor diameter ≤10 mm. Both cases have all the typical imaging findings but differ in histopathological grade. (**a**–**c**). G1-PNET (arrow), tumor diameter: 9 mm, Ki-67 labeling index: 0.8% (**d**–**f**). G2-PNET (arrow), tumor diameter: 10 mm, Ki-67 labeling index: 7.0% The acquisition phase of the CT images (**a**,**d**) is the arterial phase, and the histological images (**c**,**f**) show Ki-67 immunostaining with an original magnification 200×. PNET, pancreatic neuroendocrine tumor; CT, computed tomography.

**Table 1 diagnostics-15-02423-t001:** Clinical characteristics of 107 patients with PNEN stratified by tumor size.

	All Patients (*n* = 107)	≤10 mm (*n* = 42)	>10 mm (*n* = 65)	*p*-Value
Age (years)	60 (53–73)	67 (54–73)	58 (52–73)	0.204
Sex (male to female)	59/48	23/19	36/29	0.950
Size of the tumor (mm)	13 (8–23)	7 (5–9)	19 (14–32)	<0.001
Location of the tumor				
Pancreatic head	36 (33.6)	10 (23.8)	26 (24.3)	0.084
Pancreatic body	31 (29.0)	17 (40.5)	14 (13.1)	0.035
Pancreatic tail	40 (37.4)	15 (35.7)	25 (23.4)	0.774
Number of lesions				
Single	91 (85.0)	36 (85.7)	55 (51.4)	0.876
Multiple	16 (15.0)	6 (14.3)	10 (9.3)	
Grade (WHO, 2022) [19]				
NET G1	58 (54.2)	29 (69.0)	29 (27.1)	0.013
NET G2	40 (37.4)	13 (31.0)	27 (25.2)	0.269
NET G3	4 (3.7)	0	4 (3.7)	0.101
NEC (small cell type)	5 (4.7)	0	5 (4.7)	0.066
Non-functional	88 (82.2)	37 (88.1)	51 (47.7)	0.203
Functional Status	19 (17.8)	5 (11.9)	14 (13.1)	
Insulinoma	15 (14.0)	4 (9.5)	11 (10.3)	0.282
Gastrinoma	4 (3.7)	1 (2.4)	3 (2.8)	0.552
Hereditary status				
MEN type 1	6 (5.6)	1 (2.4)	5 (4.7)	0.244
Symptom				
Asymptomatic	65 (60.7)	30 (71.4)	35 (32.7)	0.069
Symptomatic	42 (39.3)	12 (28.6)	30 (28.0)	
EUS-FNA performed	85 (79.4)	34 (81.0)	51 (47.7)	0.756
Needle size				
19 gauge	1 (1.2)	0	1 (2.0)	
22 gauge	38 (44.7)	10 (29.4)	28 (54.9)	
25 gauge	45 (52.9)	23 (67.6)	22 (43.1)	
Unknown	1 (1.2)	1 (2.9)	0	
Type of needle				
FNA needle	68 (80.0)	30 (88.2)	38 (74.5)	
FNB needle	12 (14.1)	2 (5.9)	10 (19.6)	
Surgical resection performed	92 (86.0)	38 (90.5)	54 (83.1)	0.282

Data are expressed as number (percentage) or median (interquartile range). Abbreviations: PNEN, pancreatic neuroendocrine neoplasms; WHO, World Health Organization; NET, neuroendocrine tumor; NEC, neuroendocrine carcinoma; MEN, multiple endocrine neoplasia; EUS-FNA, endoscopic ultrasound-guided fine needle aspiration; FNB, fine needle biopsy.

**Table 2 diagnostics-15-02423-t002:** Detection rate of PNEN by EUS and contrast-enhanced CT.

	All Patients	≤10 mm	>10 mm	*p*-Value
EUS (*n* = 101)				
Lesions detected	98/101 (97.0)	36/39 (92.3)	62/62 (100)	0.027
Contrast-enhanced CT (*n* = 106)				
Lesions detected	97/106 (91.5)	35/42 (83.3)	62/64 (96.9)	0.014

Data are expressed as numbers (percentage). Abbreviations: PNEN, pancreatic neuroendocrine neoplasms; EUS, endoscopic ultrasound; CT, computed tomography.

**Table 3 diagnostics-15-02423-t003:** Comparison of imaging findings between PNEN ≤10 mm and >10 mm.

	≤10 mm (*n* = 36)	>10 mm (*n* = 61)	*p*-Value
Images			
Typical	26 (72.2)	16 (26.2)	<0.001
Atypical	10 (27.8)	45 (73.8)	
Shape			
Regular	33 (91.7)	40 (65.6)	0.004
Irregular	3 (8.3)	21 (34.4)	
Internal uniformity			
Homogeneous	35 (97.2)	34 (55.7)	<0.001
Heterogeneous	1 (2.8)	27 (44.3)	
Early enhancement			
Present	32 (88.9)	35 (57.4)	0.001
Absent	4 (11.1)	26 (42.6)	
Calcification			
Absent	35 (97.2)	49 (80.3)	0.018
Present	1 (2.8)	12 (19.7)	
Cystic degeneration			
Absent	34 (94.4)	48 (78.7)	0.038
Present	2 (5.6)	13 (21.3)	
MPD dilation			
Absent	36 (100)	47 (77.0)	0.002
Present	0 (0)	14 (23.0)	

Data are expressed as numbers (percentage). Abbreviations: PNENs, pancreatic neuroendocrine neoplasms; MPD, main pancreatic duct.

**Table 4 diagnostics-15-02423-t004:** Comparison of NET G1 and G2 imaging findings in the group with tumor diameter ≤10 mm.

	NET G1 (*n* = 25)	NET G2 (*n* = 11)	*p*-Value
Images			
Typical	18 (72.0)	8 (72.7)	0.964
Atypical	7 (28.0)	3 (27.3)	
Shape			
Regular	22 (88.0)	11 (100)	0.230
Irregular	3 (12.0)	0 (0)	
Internal uniformity			
Homogeneous	24 (96.0)	11 (100)	0.501
Heterogeneous	1 (4.0)	0 (0)	
Early enhancement			
Present	23 (92.0)	9 (81.8)	0.371
Absent	2 (8.0)	2 (18.2)	
Calcification			
Absent	24 (96.0)	11 (100)	0.501
Present	1 (4.0)	0 (0)	
Cystic degeneration			
Absent	24 (96.0)	10 (90.9)	0.539
Present	1 (4.0)	1 (9.1)	
MPD dilation			
Absent	25 (100)	11 (100)	-
Present	0 (0)	0 (0)	

Data are expressed as numbers (percentage). Abbreviations: NET, neuroendocrine tumor; MPD, main pancreatic duct.

**Table 5 diagnostics-15-02423-t005:** Comparison of imaging findings according to pathological tumor grade.

	NET G1 (*n* = 52)	NET G2 (*n* = 37)	NET G3/NEC (*n* = 8)	*p*-Value
Images				
Typical	27 (51.9)	15 (40.5)	0	0.020
Atypical	25 (48.1)	22 (59.5)	8 (100)	
Shape				
Regular	45 (86.5)	27 (73.0)	1 (12.5)	<0.001
Irregular	7 (13.5)	10 (27.0)	7 (87.5)	
Internal uniformity				
Homogeneous	44 (84.6)	24 (64.9)	1 (12.5)	<0.001
Heterogeneous	8 (15.4)	13 (35.1)	7 (87.5)	
Early enhancement				
Present	42 (80.8)	25 (67.6)	0	<0.001
Absent	10 (19.2)	12 (32.4)	8 (100)	
Calcification				
Absent	48 (92.3)	30 (81.1)	6 (75.0)	0.187
Present	4 (7.7)	7 (18.9)	2 (25.0)	
Cystic degeneration				
Absent	43 (82.7)	31 (83.8)	8 (100)	0.446
Present	9 (17.3)	6 (16.2)	0 (0)	
MPD dilation				
Absent	51 (98.1)	27 (73.0)	5 (62.5)	<0.001
Present	1 (1.9)	10 (27.0)	3 (37.5)	

Data are expressed as numbers (percentage). Abbreviations: NET, neuroendocrine tumor; NEC, neuroendocrine carcinoma; MPD, main pancreatic duct.

**Table 6 diagnostics-15-02423-t006:** Histological diagnosis accuracy of EUS-FNA for PNENs based on tumor size.

	≤10 mm (*n* = 34)	>10 mm (*n* = 51)	*p*-Value
Technical success rate	34/34 (100)	51/51 (100)	-
Rate of histopathologically diagnosed PNENs	31/34 (91.2)	44/51 (86.3)	0.492
Rate of diagnosis by histological examination	28/34 (82.4)	43/51 (84.3)	0.811
Rate of diagnosis by cytology alone	3/34 (8.8)	1/51 (2.0)	
Rate of histopathological grade diagnosed	27/31 (87.1)	41/44 (93.2)	0.372
Grade concordance rate with surgical specimens	21/24 (87.5)	18/32 (56.3)	0.012

Data are expressed as numbers (percentage). Abbreviations: EUS-FNA, endoscopic ultrasound-guided fine needle aspiration; PNENs, pancreatic neuroendocrine neoplasms.

## Data Availability

The data presented in this study are available on request from the corresponding author. The data are not publicly available due to privacy.

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
