# Peer review of "Clinicopathological Features of Small Pancreatic Neuroendocrine Neoplasms 10 mm or Smaller"

_diagnostics, 2025, doi:10.3390/diagnostics15192423_

Round 1

Reviewer 1 Report

Comments and Suggestions for Authors

The paper is interesting but it needs to be consistently improved.

1) My main concern is why NETs less than 10 mm were sampled? This is not in accordance with the current guidelines and the authors should explain why they did not just watch and see these small lesions

2) It seems the authors used either FNB or FNA needles. THis should be explained. Also, why they used sometimes FNA and sometimes FNB? And why they used different needle sizes?

3) The retrospective design and the relatively limited sample size represent limitations to the study

4) Some images would improve the quality of the manuscript

5) The authors should comment on the potential impact of sampling techniques on the diagnostic outcomes in these small lesions, citing the relevant references PMID: 36657607 and PMID: 35915956 )

6) The authors should explain which sampling technique they used and if ROSE was available at their centers

Author Response

Response to Reviewer 1

We wish to express our appreciation to the Reviewer for the insightful comments, which have helped us significantly improve the paper.

Comment 1: My main concern is why NETs less than 10 mm were sampled? This is not in accordance with the current guidelines and the authors should explain why they did not just watch and see these small lesions.

Response: We thank the reviewer for this important comment. We agree that, in accordance with current ENETS and NCCN recommendations, many nonfunctioning PNENs ≤10 mm without high-risk features can be managed with surveillance. Our cohort spans 2011 to 2024, a period during which practice patterns evolved. In the earlier years at our institution, early resection was sometimes pursued for ≤10 mm lesions when (i) imaging was atypical, (ii) patients preferred definitive diagnosis or treatment, or (iii) diagnostic certainty was prioritized. We have clarified this historical context in Methods (Participants) and now explicitly note that contemporary guidelines differ from earlier practice. We have added the following text in the Materials and Methods:

“At our institution, even small PNENs (≤10 mm) were surgically resected in selected cases. Resection was considered when imaging findings were atypical, when patients preferred definitive diagnosis or treatment, or when histological confirmation was required. During the earlier years of the study period (2011–2024), clinical guidance on the management of small PNENs was evolving, and a standardized watch-and-wait approach had not yet been widely implemented in routine practice.”

Comment 2: It seems the authors used either FNB or FNA needles. THis should be explained. Also, why they used sometimes FNA and sometimes FNB? And why they used different needle sizes?

Response: We thank the reviewer for this insightful comment. We agree that both FNA and FNB needles were used in our study, with varying needle sizes. The selection of needle type (FNA vs. FNB) and gauge was largely at the discretion of the endoscopist, based on factors such as lesion characteristics (for example, size, location, vascularity) and procedural stability. Importantly, practice also reflected temporal changes in device availability and supporting evidence. In the earlier years of the study period, FNA needles were more commonly used at our center. With the increasing availability of dedicated core biopsy needles and accumulating evidence, the use of FNB increased in the later years, particularly when preservation of tissue architecture was prioritized or when a PNEN was suspected. These points are now clarified in the Materials and Methods. We have changed the following text in the Materials and Methods:

“The choice of needle type and gauge was made at the discretion of the endoscopist.”

To

“The choice between FNA and FNB needles, as well as the needle gauge (19G or 22G or 25G), was at the discretion of the endoscopist based on lesion characteristics and procedural stability. FNA was more commonly used in the earlier years of the study period, and the use of FNB increased in the later years with broader availability of core needles and growing evidence, particularly when architectural preservation was required or when a PNEN was suspected.”

Comment 3: The retrospective design and the relatively limited sample size represent limitations to the study.

Response: We thank the reviewer for this important comment. We fully agree and have strengthened the language in the Discussion.

“This study has certain limitations. First, it was a single-center retrospective study with a relatively small sample size, which may limit the generalizability and statistical power of the findings.”

To

“This study has several important limitations. First, it was a retrospective study conducted at a single center. Second, the sample size, particularly for some subgroup analyses, is modest, which may limit external generalizability and statistical power to detect subtle differences or associations.”

Comment 4: Some images would improve the quality of the manuscript.

Response: We thank the reviewer for this valuable suggestion. In response, we have added a study flow diagram to clarify case identification, exclusions, and the final analytic cohorts (Figure 1). We have also included representative histopathological and immunohistochemical images to enhance clarity. Specifically, we added hematoxylin and eosin (H and E) stained images of both well differentiated and poorly differentiated PNENs, as well as immunohistochemical staining for chromogranin A, synaptophysin, and Ki 67 spanning G1 to NEC (Supplementary Figure S1). We have added the following text and figures in the Materials and Methods:

“Representative histopathological images are presented in Supplementary Figure S1.”

Comment 5: The authors should comment on the potential impact of sampling techniques on the diagnostic outcomes in these small lesions, citing the relevant references (PMID: 36657607 and PMID: 35915956).

Response: We thank the reviewer for this valuable suggestion. We have now added a statement in the Discussion section addressing the potential impact of different sampling techniques on diagnostic outcomes. we have added the following text in the Discussion:

“Sampling technique may influence diagnostic outcomes in small PNENs. Although our study did not stratify results by aspiration methods, prior evidence indicates that technique affects tissue adequacy and integrity. A recent network meta-analysis reported that the modified wet-suction technique yielded the highest sample adequacy and tissue integrity, though it was associated with greater blood contamination [26]. In a multicenter randomized crossover trial demonstrated that wet-suction achieved higher tissue core procurement and greater tissue integrity compared with the slow-pull technique, while diagnostic accuracy and tumor fraction were similar between the two methods [27]. These findings suggest that differences in sampling technique could partly explain variability in reported outcomes for small PNENs and should be considered in both clinical practice and future research.”

Comment 6: The authors should explain which sampling technique they used and if ROSE was available at their centers.

Response: We thank the reviewer for this important comment. As described in the Methods, both slow-pull and suction aspiration techniques were used, with selection determined by the endoscopist based on lesion characteristics and procedural stability. Because in some cases both slow-pull and wet-suction were employed within the same procedure, we did not stratify diagnostic outcomes by technique. ROSE was available at our institution throughout the study period and was performed when a cytopathologist or trained cytotechnologist was present during the procedure. We have added the following text in the Materials and Methods:

“Rapid On-Site Evaluation (ROSE) was available throughout the study period and was performed when a cytopathologist or trained cytotechnologist was present during the procedure.

Reviewer 2 Report

Comments and Suggestions for Authors

I had the opportunity to review the manuscript "Clinicopathological Features of Patients with Pancreatic Neuroendocrine Neoplasms 10 mm or Smaller in Diameter". The study is devoted to the topical topic of clinicopathological features of small pancreatic neuroendocrine neoplasms, which is important given the increasing frequency of their detection and the complexity of patient management. The topic of the study is extremely relevant for clinicians and pathologists working in the field of pancreatic oncology. The authors collected a sufficient number of observed cases, which ensures statistical power and reliability of the results obtained. Statistical processing of the data was carried out adequately.
Despite the relevance and high-quality methodology, I have several significant comments that the authors should consider to improve the manuscript.

- Outdated classification and staging: This is the most important comment. The paper uses the 2019 World Health Organization (WHO) classification, while from 2024, the updated 2024 WHO classification and the corresponding grading approach should be used in clinical practice and scientific publications. Authors should completely revise the "Materials and Methods" and "Results" sections, as well as the corresponding tables and figures, to comply with the current WHO recommendations.
- The manuscript devoted to clinicopathological features does not contain visualization of histopathological studies. It is strongly recommended to add:

a) Photomicrographs (hematoxylin-eosin staining) illustrating the morphological features of the tumors.

b) Immunohistochemical verification images demonstrating positive staining for neuroendocrine markers (e.g., chromogranin A, synaptophysin).

c) Photographs of the Ki-67 study, which allow assessing the proliferation index, which is critical for determining the degree of malignancy (grading).
- The title "Clinicopathological Features of Patients with Pancreatic Neuroendocrine Neoplasms 10 mm or Smaller in Diameter" does not fully reflect the content of the work. I recommend that the authors revise the title so that it better reflects the focus of the study, for example, "Clinicopathological Features of Small Pancreatic Neuroendocrine Neoplasms" or another option that more accurately describes the content.
- The conclusions in the abstract need significant improvement. The abstract should be more informative and concisely reflect the key results of the study, emphasizing their clinical significance.
Overall, the manuscript has potential, but in its current form it requires refinement. The above comments regarding the outdated classification and the lack of pathohistological images are important.
I await a revised version of the manuscript that takes these comments into account, as their correction will significantly enhance the scientific value and clinical relevance of the work.

Author Response

Response to Reviewer 2

We wish to express our appreciation to the Reviewer for the insightful comments, which have helped us significantly improve the paper.

Comment 1: Outdated classification and staging: This is the most important comment. The paper uses the 2019 World Health Organization (WHO) classification, while from 2024, the updated 2024 WHO classification and the corresponding grading approach should be used in clinical practice and scientific publications. Authors should completely revise the "Materials and Methods" and "Results" sections, as well as the corresponding tables and figures, to comply with the current WHO recommendations.

Response: We sincerely thank the reviewer for pointing out this important issue. We fully acknowledge that the 2022 WHO classification (published as part of the WHO 5th edition series and expected to be adopted in clinical and scientific practice from 2024 onwards) represents the most up-to-date standard for the classification and grading of pancreatic neuroendocrine neoplasms (PNENs). We have now revised the classification and grading system used throughout the manuscript to fully align with the 2022 WHO criteria. We have changed the following text in the Materials and Methods and Table 1:

“2019”

To

“2022”

Comment 2: The manuscript devoted to clinicopathological features does not contain visualization of histopathological studies. It is strongly recommended to add:

  1. a) Photomicrographs (hematoxylin-eosin staining) illustrating the morphological features of the tumors.
  2. b) Immunohistochemical verification images demonstrating positive staining for neuroendocrine markers (e.g., chromogranin A, synaptophysin).
  3. c) Photographs of the Ki-67 study, which allow assessing the proliferation index, which is critical for determining the degree of malignancy (grading).

Response: We sincerely thank the reviewer for this valuable suggestion. In accordance with the suggestion, we have added representative histopathological and immunohistochemical images to enhance clarity. Specifically, we added hematoxylin and eosin (H and E) stained images of both well differentiated and poorly differentiated PNENs, as well as immunohistochemical staining for chromogranin A, synaptophysin, and Ki 67 spanning G1 to NEC (Supplementary Figure S1).

Comment 3: The title "Clinicopathological Features of Patients with Pancreatic Neuroendocrine Neoplasms 10 mm or Smaller in Diameter" does not fully reflect the content of the work. I recommend that the authors revise the title so that it better reflects the focus of the study, for example, "Clinicopathological Features of Small Pancreatic Neuroendocrine Neoplasms" or another option that more accurately describes the content.

Response: We thank the reviewer for the helpful suggestion regarding the title. To better reflect the focus of the study while incorporating the reviewer’s wording, we have revised the title to “Clinicopathological Features of Small Pancreatic Neuroendocrine Neoplasms 10 mm or Smaller.” This preserves the emphasis on small PNENs and clarifies that “small” refers to lesions 10 mm or smaller in diameter.

Comment 4: The conclusions in the abstract need significant improvement. The abstract should be more informative and concisely reflect the key results of the study, emphasizing their clinical significance.

Response: We sincerely thank the reviewer for this valuable comment. We agree that the original abstract conclusion did not sufficiently emphasize clinical relevance. Accordingly, we revised the conclusion to more clearly reflect the key results and their implications for care, highlighting the limitations of imaging alone for grade assessment and the diagnostic contribution of EUS-FNA in guiding management of small PNENs. We have changed the following text in the Abstract:

Conclusions: Although small PNENs (≤10 mm) frequently exhibit typical imaging characteristics, tumor grade cannot be reliably determined by imaging alone. EUS-FNA demonstrated high diagnostic accuracy, supporting the importance of pathological evaluation in the management of small PNENs.”

To

Conclusions: Conclusions: In small PNENs (≤10 mm), imaging features are often typical but do not reliably determine tumor grade. In our cohort, EUS-FNA showed high diagnostic accuracy and provided essential pathological information to guide management, including the choice between surveillance and surgery.”

Reviewer 3 Report

Comments and Suggestions for Authors

Comments:

1. The Introduction should be tightened to avoid unnecessary background and provide a sharper focus. After briefly contextualizing the clinical or scientific problem, it should end with a clear statement of the hypothesis, the primary aim, and the key expected contribution of the study.

2. I would suggest to the authors that all statistically significant results reported in the tables should also be presented graphically, for example using bar charts. This would make the findings more accessible, highlight key differences more clearly, and improve the overall readability of the results section.

3. The composite “typical” definition requiring all six criteria is overly strict and may bias results toward larger or higher-grade lesions. The rationale for this choice should be explained, and a sensitivity analysis is needed to show whether findings hold when each feature is examined individually or when a cumulative score (0–6) is modeled—preferably with ordinal logistic regression adjusted for size. This would clarify whether the observed associations reflect the strict definition itself or a more general graded relationship.

4. Complications of EUS-FNA are not reported and should be detailed to contextualize safety. Please provide per-patient and per-procedure adverse-event rates with severity grading (e.g., Clavien–Dindo or ASGE)

5. The study’s statistical strength is not addressed, and this should be clarified. I recommend adding a statement on how the power of the study was determined. Ideally, the Methods should specify whether an a priori sample size calculation was performed based on the expected effect size, alpha, and power. If this was not done, the authors should provide at least a post hoc power or precision analysis, showing the smallest effect size the current sample could reliably detect with 80% power at a 5% significance level.

6. The manuscript would be strengthened by a clearer description of the study population. I recommend that the authors explicitly define the inclusion and exclusion criteria in the Methods section. In addition, a patient flow diagram should be provided, detailing the total number of patients screened, the reasons for exclusion, and the final number included in the analysis. This PRISMA-style flow chart will improve transparency. 

7. The definition of MPD dilatation needs clearer justification. Using a uniform cutoff of ≥3 mm across the entire pancreas may be problematic, since physiological duct size differs by segment, with the head/neck region often larger than the tail. Please explain the rationale for applying a single threshold. 

8. Functional status (insulinoma/gastrinoma) and MEN1 may confound imaging features and management pathways, potentially biasing associations. Please either adjust for these covariates in multivariable models or add sensitivity analyses excluding functioning tumors and/or MEN1 cases to confirm that the main findings are not driven by these subgroups.

9. Please ensure that Figures 1 and 2 are prepared in high resolution, with histological images shown in full color and including scale bars with units, while CT images should indicate the acquisition phase directly in the legend. In both figure types, arrows or arrowheads should be added to highlight the relevant lesions or structures. 

10. Numerous typos impede readability (e.g., “Englis text,” “neuroendcrine,” spacing/line breaks). A professional edit is needed. 

11. I suggest creating a dedicated Future Directions section to outline specific next steps, including how the findings could be incorporated into clinical practice. It would also be useful to include a concise Key Clinical Box that distills the main take-home messages for practitioners. Finally, a summary figure or table synthesizing the study design, main predictors, and clinical outcomes.

Author Response

Response to Reviewer 3

We wish to express our appreciation to the Reviewer for the insightful comments, which have helped us significantly improve the paper.

Comment 1: The Introduction should be tightened to avoid unnecessary background and provide a sharper focus. After briefly contextualizing the clinical or scientific problem, it should end with a clear statement of the hypothesis, the primary aim, and the key expected contribution of the study.

Response: We thank the reviewer for this helpful suggestion. In accordance with the Reviewer’s comment, we have changed text in the Introduction:

“In recent years, advances in abdominal ultrasound, computed tomography (CT), magnetic resonance imaging (MRI), and endoscopic ultrasound (EUS) have increased the incidental detection of small pancreatic neuroendocrine neoplasms (PNENs) [1,2]. Cur-rent clinical guidelines recommend treatment strategies for non-functional PNENs (NF-PNENs) mainly according to tumor size, with >20 mm generally used as the thresh-old for surgical resection because larger tumors are associated with higher malignant potential [3,4]. However, the optimal management of small NF-PNENs remains controversial [5–8].

For incidentally discovered NF-PNENs ≤10 mm, active surveillance is often considered preferable to surgery, given the risks of perioperative complications and long-term sequelae [9–11]. Nonetheless, several studies have shown that even tumors ≤10 mm may harbor malignant potential, including lymph node metastasis, particularly when atypical imaging features are present or when the tumor is classified as G2 [12]. These findings suggest that tumor size alone is not sufficient for reliable prognostic assessment. Instead, accurate pathological grading, especially based on the Ki-67 index, is essential to guide treatment decisions.

Endoscopic ultrasound-guided fine needle aspiration (EUS-FNA) is the most widely used method for preoperative pathological diagnosis of PNENs, but its diagnostic accuracy in very small tumors (≤10 mm) remains unclear. Concerns have been raised regarding technical feasibility, sample adequacy, and the reliability of grading in such small lesions [13–15]. Moreover, few studies have directly compared imaging findings with histopathological grade in this subgroup.

We hypothesized that EUS-FNA provides more accurate pathological grading for small PNENs (≤10 mm) than for larger tumors, and that imaging findings alone cannot reliably distinguish between G1 and G2 tumors of this size. The primary aim of this study was to evaluate the diagnostic accuracy of EUS-FNA and to compare imaging and patho-logical features according to tumor size. By clarifying these aspects, our study contributes to optimizing clinical decision-making regarding surveillance versus surgical resection for small PNENs.”

Comment 2: I would suggest to the authors that all statistically significant results reported in the tables should also be presented graphically, for example using bar charts. This would make the findings more accessible, highlight key differences more clearly, and improve the overall readability of the results section.

 Response: Thank you for this helpful suggestion. To enhance readability without duplicating tabulated numbers, we added a concise supplementary figure that visualizes the size-stratified imaging features from Table 3 as bar charts with 95 percent confidence intervals and n labels. We have added the following text in the Results:

“Size-stratified imaging features are visualized in Supplementary Figure S2, complementing Table 3 and facilitating comparison between ≤10 mm and >10 mm.”

Comment 3: The composite “typical” definition requiring all six criteria is overly strict and may bias results toward larger or higher-grade lesions. The rationale for this choice should be explained, and a sensitivity analysis is needed to show whether findings hold when each feature is examined individually or when a cumulative score (0–6) is modeled—preferably with ordinal logistic regression adjusted for size. This would clarify whether the observed associations reflect the strict definition itself or a more general graded relationship.

Response: Thank you for this thoughtful comment. To examine whether our findings depend on the strict all-criteria definition or reflect a graded relationship, we modeled a cumulative atypicality score from 0 to 6 (higher indicates more atypical) using a proportional odds ordinal logistic regression in JMP, with lesion size entered in 10 mm units. Lesion size was associated with higher atypicality categories; the odds ratio per 10 mm increase was 0.498 (95% CI 0.374 to 0.646; P < 0.001), indicating that larger lesions tended to exhibit more atypical features. Results were similar after collapsing sparse upper categories. These analyses support a graded relationship rather than an artifact of the strict 6-of-6 threshold.

We have added the following text in the Materials and Methods:

“An atypicality score from 0 to 6 (higher values indicate more atypical appearance) was constructed by summing six predefined imaging features. The score was modeled as an ordinal outcome using proportional odds logistic regression. Lesion size was re-scaled in 10 mm units, and odds ratios per 10 mm increase with 95 percent confidence intervals (CIs) were reported.”

In addition, we have added the following text in the Results:

“Lesion size was associated with higher atypicality categories. The odds ratio per 10 mm increase was 0.498 (95% CI 0.374–0.646; P < 0.001), indicating a shift toward more atypical features with increasing size.”

Comment 4: Complications of EUS-FNA are not reported and should be detailed to contextualize safety. Please provide per-patient and per-procedure adverse-event rates with severity grading (e.g., Clavien–Dindo or ASGE).

Response: Thank you for highlighting the need to contextualize safety. In this cohort, no procedure-related adverse events occurred among the 85 patients who underwent EUS-FNA, corresponding to a 0 percent adverse-event rate per patient and per procedure. Adverse events were prospectively monitored and retrospectively confirmed from clinical records, laboratory data, and imaging. Severity was to be graded according to the ASGE lexicon for endoscopic adverse events; as no events were observed, no grading applied. We have added the following text in the Materials and Methods, and Results.

“Adverse events related to EUS-FNA were monitored within 7 days using medical rec-ords, laboratory results, and imaging, and severity was to be classified according to the American Society for Gastrointestinal Endoscopy lexicon for endoscopic adverse events [18].” 

And

“No procedure-related complications occurred among the 85 patients who underwent EUS-FNA, yielding per-patient and per-procedure adverse event rates of 0%; no severity grading applied.”

Comment 5: The study’s statistical strength is not addressed, and this should be clarified. I recommend adding a statement on how the power of the study was determined. Ideally, the Methods should specify whether an a priori sample size calculation was performed based on the expected effect size, alpha, and power. If this was not done, the authors should provide at least a post hoc power or precision analysis, showing the smallest effect size the current sample could reliably detect with 80% power at a 5% significance level.

Response: Thank you for this important point. The primary outcome was the between-group comparison by size category (≤10 mm vs >10 mm). Because this was a retrospective consecutive cohort, no a priori sample size calculation was performed. To contextualize precision, the Methods now report a post hoc minimum detectable absolute difference at alpha 0.05 and power 0.80 using the observed group sizes (≤10 mm, n=42; >10 mm, n=65). Depending on the assumed baseline proportion in the reference group, the minimum detectable difference is approximately 24 to 27 percentage points for baseline rates of 20 to 40 percent.

We have added the following text in the Materials and Methods:

“For the primary comparison between ≤10 mm (n=42) and >10 mm (n=65), a post hoc minimum detectable absolute difference at alpha = 0.05 and power = 0.80 was computed, yielding approximately 24–27 percentage points across baseline proportions of 20 to 40 percent in the reference group.”

Comment 6: The manuscript would be strengthened by a clearer description of the study population. I recommend that the authors explicitly define the inclusion and exclusion criteria in the Methods section. In addition, a patient flow diagram should be provided, detailing the total number of patients screened, the reasons for exclusion, and the final number included in the analysis. This PRISMA-style flow chart will improve transparency. 

Response: Thank you for this valuable suggestion. The Materials and Methods section has been revised to explicitly define the inclusion and exclusion criteria. We included patients with pancreatic neuroendocrine neoplasms who had histopathological confirmation by EUS-FNA or surgical resection; patients without pathological evaluation were excluded. In addition, a PRISMA-style patient flow diagram has been added as Figure 1 to depict the selection process, including the total number assessed, reasons for exclusion, and the final cohort included. During the study period, 109 patients were assessed, 2 were excluded for lack of histopathological confirmation, and 107 were included in the analysis (≤10 mm, n=42; >10 mm, n=65). We believe these revisions improve transparency and clarity.

Comment 7: The definition of MPD dilatation needs clearer justification. Using a uniform cutoff of ≥3 mm across the entire pancreas may be problematic, since physiological duct size differs by segment, with the head/neck region often larger than the tail. Please explain the rationale for applying a single threshold. 

Response: Thank you for this thoughtful comment. Physiological MPD caliber varies by segment; however, to ensure consistent and objective assessment across CT and EUS in this retrospective cohort, tumor-related MPD dilatation was prespecified as ≥3 mm. This single threshold follows prior clinical literature that has adopted ≥3 mm as a practical criterion for abnormal ductal enlargement in the PNEN context (PMID: 31718651, 30451795). Operationally, dilatation was assessed at the maximally distended ductal segment immediately proximal to the index lesion.

We have changed the following text in the Materials and Methods:

“MPD dilation: maximal MPD diameter ≥3 mm distal to the lesion.”

to

“MPD dilation: maximal MPD diameter ≥3 mm proximal to the index lesion on CT or EUS [16,17].”

Comment 8: Functional status (insulinoma/gastrinoma) and MEN1 may confound imaging features and management pathways, potentially biasing associations. Please either adjust for these covariates in multivariable models or add sensitivity analyses excluding functioning tumors and/or MEN1 cases to confirm that the main findings are not driven by these subgroups.

Response: Thank you for this important point. To assess whether functioning tumors and MEN1 could bias the associations, we conducted sensitivity analyses excluding functioning tumors and MEN1 cases. Details are provided in Supplementary Table S1, and we have added the following text in the Results:

“In sensitivity analyses excluding functioning tumors and/or MEN1 cases, the direction and magnitude of the associations were materially unchanged, with significant between-group differences persisting for most imaging features (Supplementary Table S1).”

Comment 9: Please ensure that Figures 1 and 2 are prepared in high resolution, with histological images shown in full color and including scale bars with units, while CT images should indicate the acquisition phase directly in the legend. In both figure types, arrows or arrowheads should be added to highlight the relevant lesions or structures. 

Response: Thank you for these helpful suggestions. We have revised the figures and legends accordingly. Because a calibrated scale bar could not be added to the archived images, we report the original magnification (×200) in the legend as an alternative.

“Figure 2. Typical (a, b) and atypical (c–h) imaging findings for pancreatic neuroendocrine neoplasms on contrast-enhanced computed tomography (CT) and endoscopic ultrasonography (EUS)

  1. Arterial-phase CT showing a tumor with early enhancement (arrow).
  2. EUS showing a tumor with smooth margins and homogeneous internal echotexture (arrow).
  3. EUS showing a tumor with irregular margins (arrow).
  4. EUS showing a tumor with internal heterogeneity (arrow).
  5. Arterial-phase CT showing a tumor without early enhancement (arrow).
  6. CT showing intratumoral calcifications (arrowhead) within the tumor (arrow).
  7. EUS showing a tumor (arrow) with cystic degeneration (arrowhead).
  8. CT showing a tumor (arrow) with main pancreatic duct dilatation (arrowhead).”

“Figure 3. Two typical imaging cases in the group with tumor diameter ≤10 mm

Both cases have all the typical imaging findings but differ in histopathological grade.

a–c. G1-PNET (arrow), tumor diameter: 9 mm, Ki-67 labeling index: 0.8%

d–f. G2-PNET (arrow), tumor diameter: 10 mm, Ki-67 labeling index: 7.0%

The acquisition phase of the CT images (a, d) is the arterial phase, and the histological images (c, f)

show Ki-67 immunostaining with an original magnification ×200.

PNET, pancreatic neuroendocrine tumor; CT, computed tomography”

Comment 10: Numerous typos impede readability (e.g., “Englis text,” “neuroendcrine,” spacing/line breaks). A professional edit is needed. 

Response: We sincerely thank the reviewer for pointing out the typographical and formatting issues that affected the readability of our manuscript. We have thoroughly revised the entire manuscript to correct all typographical errors, including spelling mistakes and inconsistencies in spacing and line breaks.

Comment 11: I suggest creating a dedicated Future Directions section to outline specific next steps, including how the findings could be incorporated into clinical practice. It would also be useful to include a concise Key Clinical Box that distills the main take-home messages for practitioners. Finally, a summary figure or table synthesizing the study design, main predictors, and clinical outcomes.

Response: Thank you for the constructive suggestion. To improve clinical readability without expanding the manuscript beyond the journal’s customary format, a concise paragraph has been added at the end of the Discussion to outline future directions and how the findings may be incorporated into practice.

“Future directions and clinical implications:

The present findings indicate that imaging typicality alone is insufficient for reliable grading of very small PNENs and that pathological assessment obtained by EUS-FNA contributes meaningfully to size-stratified management, including the choice between surveillance and surgery. In clinical pathways, EUS-FNA may be prioritized for lesions 10 mm or smaller when atypical imaging features are present or when grade determination would alter treatment. Future work should validate the graded atypicality framework in multicenter cohorts using standardized sampling techniques and prespecified analytic plans, and evaluate prospective triggers for surveillance or intervention anchored to pathological grade. These steps could help translate the current evidence into practical algorithms while minimizing both overtreatment and undertreatment.”

Round 2

Reviewer 1 Report

Comments and Suggestions for Authors

The revised manuscript is OK

Reviewer 3 Report

Comments and Suggestions for Authors

The authors have addressed all of my comments, and the manuscript can be accepted for publication.